# Following Evidence-Based Recommendations for Perioperative Pain Management after Cesarean Section Is Associated with Better Pain-Related Outcomes: Analysis of Registry Data [note 1]

**DOI:** 10.3390/jcm12020676

**Published:** 2023-01-14

**Authors:** Ruth Zaslansky, Philipp Baumbach, Ruth Edry, Sean Chetty, Lim Siu Min, Isabelle Schaub, Jorge Jimenez Cruz, Winfried Meissner, Ulrike M. Stamer

**Affiliations:** 1Department of Anesthesiology and Intensive Care Medicine, University Hospital Jena, 07747 Jena, Germany; 2Acute Pain Service, Department of Anesthesiology, Rambam Health Care Campus, Haifa 3109601, Israel; 3Department of Anaesthesiology& Critical Care, Faculty of Medicine and Health Sciences, Stellenbosch University, Cape Town 7500, South Africa; 4Department of Anesthesiology, Faculty of Medicine, University of Malaya, Kuala Lumpur 50603, Malaysia; 5Department of Anesthesiology and Pain Clinic, Clinique St Jean, 1000 Brussels, Belgium; 6Department of Obstetrics and Gynecology, Bonn University Hospital, 53127 Bonn, Germany; 7Department of Anaesthesiology and Pain Medicine, Inselspital, Bern University Hospital, University of Bern, 3010 Bern, Switzerland

**Keywords:** anti-inflammatory agents non-steroidal, caesarean section, pain, patient reported outcomes, registry

## Abstract

Women who have had a Cesarean Section (CS) frequently report severe pain and pain-related interference. One reason for insufficient pain treatment might be inconsistent implementation of evidence-based guidelines. We assessed the association between implementing three elements of care recommended by guidelines for postoperative pain management and pain-related patient-reported outcomes (PROs) in women after CS. The analysis relied on an anonymized dataset of women undergoing CS, retrieved from PAIN OUT. PAIN OUT, an international perioperative pain registry, provides clinicians with treatment assessment methodology and tools for patients to assess multi-dimensional pain-related PROs on the first postoperative day. We examined whether the care included [i] regional anesthesia with a neuraxial opioid OR general anesthesia with wound infiltration or a Transvesus Abdominis Plane block; [ii] at least one non-opioid analgesic at the full daily dose; and [iii] pain assessment and recording. Credit for care was given only if all three elements were administered (= “full”); otherwise, it was “incomplete”. A “Pain Composite Score-total” (PCS_total_), evaluating outcomes of pain intensity, pain-related interference with function, and side-effects, was the primary endpoint in the total cohort (women receiving GA and/or RA) or a sub-group of women with RA only. Data from 5182 women was analyzed. “Full” care was administered to 20% of women in the total cohort and to 21% in the RA sub-group. In both groups, the PCS_total_ was significantly lower compared to “incomplete” care (*p* < 0.001); this was a small-to-moderate effect size. Administering all three elements of care was associated with better pain-related outcomes after CS. These should be straightforward and inexpensive for integration into routine care after CS. However, even in this group, a high proportion of women reported poor outcomes, indicating that additional work needs to be carried out to close the evidence-practice gap so that women who have undergone CS can be comfortable when caring for themselves and their newborn.

## 1. Introduction

Cesarean section (CS) is the most common major surgical procedure in women of childbearing age [1]. Multi-center observational studies demonstrate that the incidence of severe pain after CS is high, even when compared with other gynecologic surgical procedures [2,3]. It is well described that after CS, moderate-to-severe post-operative pain and interference with function are associated with short- and long-term negative consequences for mother and child [4,5,6].

Offering care that is in line with evidence-based clinical practice guidelines is regarded as a core activity for providing quality healthcare, and when practiced, patient outcomes tend to improve [7,8,9]. Inconsistent implementation of evidence-based recommendations may be a contributing factor to the poor pain-related outcomes observed in women after CS [3,10].

The aim of this study was to evaluate whether receiving care for perioperative pain that follows recommended practices would be associated with improved pain-related patient-reported outcomes (PROs). We assessed this in a cohort of women who underwent CS and were cared for in the regular clinical routine. We hypothesized that: (1) women undergoing CS do not generally receive care that is in line with recommended practice; and (2) receiving such care would be associated with better PROs.

## 2. Methods

### 2.1. Study Design and Setting

The findings presented here rely on the analysis of an anonymized dataset of women who underwent CS and whose data was retrieved from the PAIN OUT registry. PAIN OUT (www.pain-out.eu) is a quality improvement and research network focusing on perioperative pain management (ClinicalTrials.gov NCT02083835). All collaborators obtained permission from their local ethics committee to take part in the registry. The PAIN OUT methodology for auditing perioperative pain on the first postoperative day (POD1) has been described in detail [11,12] and briefly below.

Women could be enrolled in PAIN OUT if they fulfilled the following inclusion criteria: (1) ≥18 years old; (2) being on the first postoperative day and returning to the obstetrics ward from surgery for at least six hours; and (3) consenting to take part in a survey in which they were asked to fill out a questionnaire assessing pain-related outcomes related to their surgery. Consent could be written or oral, depending on the requirements of the local ethics committee. Women were approached once by a trained surveyor on the first day after surgery.

As this was an international cohort and we had no prior information as to which methods of anesthesia were used in different countries, we included all women, regardless of the method of anesthesia they received (=total cohort). As regional anesthesia is considered the gold standard, we also created a sub-group consisting of women who received this form of anesthesia only (=RA sub-group).

### 2.2. Data Collection for Each Woman Involved Two Questionnaires Addressing

(1)Demographic characteristics, anesthesia, and surgery data include age, country of birth, type of CS (International Classification of Disease Procedure Codes, [ICD9] Code 74.x), anesthetic technique (general anesthesia, [GA]; regional anesthesia, [RA]: spinal, epidural, combined spinal epidural; GA and RA [=combined]), duration of surgery, and medications for pain administered intra-operatively and on the obstetric ward. Lastly, whether there was a record of post-operative assessment made by the nursing or medical staff.(2)Pain-related PROs using the International Pain Outcomes Questionnaire [11]. The questionnaire consists of 13 questions evaluating four outcome domains, all in relation to the time since surgery: (a) intensity of pain (worst, least pain, time spent in severe pain); (b) interference of pain with activities (changing position in bed, taking a deep breath or coughing, sleep) and with emotions (anxiety and helplessness); (c) side effects (nausea, drowsiness, itch, dizziness); and (d) perception of care (would have liked more pain treatment than received, satisfied with results of pain treatment). Most PROs are scored using a 0–10 numerical rating scale (NRS), and “I would have liked more treatment” is dichotomous (Yes/No). Lastly, women were asked about the existence and severity of pain lasting for at least 3 months before surgery. The questionnaire’s psychometric properties have been validated. The questionnaire has been translated into 29 languages using a standardized methodology. The questionnaire can be downloaded from the PAIN OUT website.

In each hospital, study surveyors—medical or nursing students, nurses, or anesthesia residents not involved in these patients’ care—underwent training for recruiting women and abstracting the demographic and clinical data from patients’ charts. Study surveyors then entered the data into a web-based, password-secure portal where each dataset was given a unique, anonymous code. There is no link between this code, the patient’s name, or the medical record from which the data was obtained. The database is hosted and maintained at the Jena University Hospital, Germany.

### 2.3. Elements of Perioperative Pain Care That Were Evaluated

We reviewed the literature, including guidelines on perioperative pain management in CS and surgery in general [5,13,14,15,16,17,18,19], that were published before or during the study’s time period. We relied largely on the 2014 version of the Procedure-Specific Pain Management (PROSPECT) guidelines for CS, which included studies from 1966 and up until 2014. PROSPECT guidelines follow a comprehensive, procedure-specific, systematic review of the literature [20]. We briefly summarized the evidence we used in Appendix A. From this, we selected three elements of perioperative pain care that were common across the resources.
**The three elements of care included:****Intra-operative phase:**Element 1:

If CS was carried out under RA, a neuraxial opioid was given.

If GA was used, the surgical wound was infiltrated with a local anesthetic or a TAP (Transversus Abdominis Plane) block was performed.**Post-operative phase:**Element 2 (for all women): A full daily dose (including intra-operatively) of a non-opioid analgesic (paracetamol or non-steroidal anti-inflammatory drug [NSAID] or metamizol [a non-opioid analgesic of choice in some of the participating countries]) was administered.

Element 3 (for all women): A member of staff assessed and recorded the patient’s pain in her chart.

Four different **care groups spanning the different perioperative treatment phases** were then created:(1)**“Full peri-operative care” (= “full” care):** if all three elements were administered during the intra- and post-operative phases, we regarded this as treatment **conforming to the recommendations**.(2)**“Intra-operative care”:** if only the ‘intra-operative phase’ elements were administered.(3)**“Post-operative care”:** if only the recovery and ward elements were administered.(4)**“Incomplete care”:** if at least one element was missing from each of the two treatment phases, above.

### 2.4. A Pain Composite Score

With acute pain being a multi-dimensional experience, patients should be assessed for several outcome domains such as intensity, function (physical or emotional), and side effects [21]. With the International Pain Outcomes Questionnaire (IPO) [11], we are able to follow these recommendations. However, recommendations are not available as to which of these domains, or a single variable, is clinically most relevant. A composite score can be an attractive option as it allows the creation of a single variable that offers a global view of the pain experience. The Pain Composite Score-total (PCS_total_) was created by averaging the continuous items from the pain intensity, pain interference, and side effects domains of the IPO. We also created sub-scores for all intensity (PCS_intensity_), interference (PCS_interference_), and side effects (PCS_side_) variables. Higher values of the PCS and sub-scores indicate worse outcomes. As a composite score, the PCS does not intuitively reflect any of the individual items from which it is composed. We, thus, also provide findings from the individual PROs.

### 2.5. Study Outcomes

The **primary endpoint** was the Pain Composite Score—total (PCS_total_)—in the “full” care group versus the “incomplete” group in the (1) total cohort and (2) RA sub-group. Secondary endpoints comprised a similar analysis for the PCS sub-scores.

### 2.6. Statistical Analysis

A data set was considered valid if the patient inclusion criteria were met, and if it included information about the three elements of care (see Section 2.3) and if it included readings for the continuous PROs in the International Pain Outcomes Questionnaire. A ward was included in the analysis if it contributed ≥30 valid data sets.

We report absolute frequencies and percentages for dichotomous variables and medians and first and third quartiles for categorical variables.

For the continuous PROs, we used ABC-Analysis [22] to compute dichotomized, pre-specified thresholds. This technique divides patient ratings into three subsets based on statistically valid definitions of thresholds (AB- and BC-limits), along with categories often used in the pain literature for a sensation that is “severe” (A), “moderate” (B), and “mild-none” (C) [23]. We used variable-specific AB-limits as cut-offs and reported the percentage of patients with “A” ratings [24].

To determine whether a full daily dose of a non-opioid analgesic was given, we calculated cumulative doses for each non-opioid (intra- and post-operative), expressed in percent of the maximum recommended dose, and adjusted for the time after surgery at which the questionnaire was filled out. Appendix A contains reference doses and calculation examples.

We used linear mixed models to evaluate the effect of implementing the care protocols, with the Pain Composite Score (total and sub-scores) as the dependent variables. Independent variables included the care groups, anesthetic technique, pre-existing chronic pain, age, opioid administration on the ward, and income level of the country. These variables are listed in some studies as having an effect on the perception of post-operative pain [25]. Random effects for every ward were also included. We used a similar regression model for patients in the RA group, including neuraxial morphine as an additional independent variable. A sensitivity analysis excluded one hospital with a high patient number. For all models, we obtained estimated marginal means, including 95% confidence intervals, and used contrasts to compare between the different care groups. *p*-values for multiple comparisons were adjusted using the Bonferroni-Holm method. *p*-values <0.05 were considered statistically significant. Due to the large sample sizes in this study, it is possible to achieve statistical significance in situations where the observed differences are clinically meaningless. In such cases, effect size provides a better basis for statistical inference. We also obtained regression models with z-standardized PCS to evaluate effect sizes. Here, contrasts were expressed as differences in the standard deviations of the PCS. We interpret the standardized regression coefficients in terms of Cohen’s d, with coefficients of ≥ 0.2/≥ 0.5/≥ 0.8 as small, medium, and large effect sizes, respectively [26]. For the analysis, we used R (version 3.6.3, Vienna, Austria [27]) and RStudio (version 1.2.5003, RStudio Inc., Boston, MA, USA [28]). We used all available data in the database and did not carry out a sample size calculation. We followed the STROBE guidelines [29] for preparing this manuscript.

## 3. Results

### 3.1. Sample Description

Of the 7620 women after CS enrolled in Pain OUT between 2010 and 2020, 5182 women from 24 obstetric wards in 21 hospitals and 15 countries qualified for inclusion (Figure 1). Of these, 40% (2090) of the women were treated in 13 obstetric wards in high-income countries, and 60% (3092) were cared for in 11 obstetric wards in middle-income (10 wards) and low-income (1 ward) countries (see Appendix A). The most common ICD-9 codes were 74.1 (low cervical CS) and 74.0 (classical CS), which accounted for 63.2% (3271) and 28.3% (1466) of cases, respectively. Within the RA group, 91.3% (4041) of women received spinal anesthesia, and 12.3% (545) received epidural anesthesia (see Appendix A). Patient characteristics, including reports of chronic pain before surgery, medications administered on the ward, and care groups, are described in Table 1.

### 3.2. Analgesics Administered

#### 3.2.1. Neuraxial Opioids

Of the neuraxial opioids, fentanyl was administered to 63% of women, alone or in combination with morphine. Morphine or sufentanil, as sole opioids, were administered to 18% and 10% of women, respectively (Table 2).

#### 3.2.2. Non-Opioid Analgesics

On the ward, 87% (5182) of women received at least one non-opioid analgesic. Paracetamol or an NSAID, as sole medications, were administered to 23.2% (n = 1201) and 18.8% (n = 976) of women, respectively. Both paracetamol and NSAIDs were administered to 38% (n = 1982) of women. All three classes (paracetamol, NSAIDs, and metamizole) were administered to 3% (n = 156) of women. However, 13% of women did not receive any non-opioid medication. When a non-opioid was administered, 80% of the cohort did not receive a full daily dose of the respective medication. The types, total daily doses, and routes of administration of opioids administered on the ward are listed in Table 1.

### 3.3. Perioperative Care Groups

“Full” care was given to 19.7% of women, whereas 62.7% were allocated to the “Incomplete” care group. “Intra-operative” and “post-operative” care was given to 3.7% and 13.9% of women, respectively (Table 1). In the RA and GA groups, 21.3% and 8.6% of women received “full” care, respectively.

### 3.4. PROs for the Total Cohort

The PROs in the “full” versus “incomplete” care groups are shown in Figure 2. Detailed descriptive statistics are listed in Appendix A. As examples, time in severe pain ≥ 50% on POD1 was reported by 27.5% versus 54% of women receiving “full” versus “incomplete” care, respectively. Anxiety ≥ 4/10 and helplessness ≥ 4/10 due to pain were reported by 27–30% versus 50% of women in “full” vs. “incomplete” care. Satisfaction ≤ 6/10 was reported by 14% versus 35% of women in “full” versus “incomplete” care. “I would have wished to receive more pain treatment” was reported by 25% versus 46% in the “full” versus “incomplete” groups.

The differences between “worst pain” and pain-related interference symptoms are minimal.

### 3.5. Regression Models

The results of the multi-level regression models for the total cohort and RA sub-group are shown in Table 3 (A) and (B), respectively. The descriptive statistics for the PCS_total_ and the sub-scores are found in Supplementary S2 Section S1.

#### 3.5.1. Primary Endpoint

After controlling for all covariates, the PCS_total_ in women receiving “full” care was significantly lower compared to women with “incomplete” care. In both the total cohort and the RA sub-group, the z-standardized PCS_total_ for women receiving “full” care was 0.36 standard deviations lower compared to women with “incomplete” care (see Table 3A,B, Figure 3A,B, and Appendix A). These are small-to- medium effect sizes.

The PCS _total_ in the “intra-operative” care group was also lower compared to “incomplete” care. It was 0.27 standard deviations lower for the total cohort and 0.24 for the RA cohort. A small effect size.

Results were similar in a sensitivity analysis excluding one hospital with a high patient number (see Appendix A).

#### 3.5.2. Secondary Endpoints

In the total cohort, women in the “full” care group showed significantly lower (i.e., better) values in the PCS_intensity_ (−0.18 SDs), PCS_interference_ (−0.35 SDs, small effect size), and PCS_side_ (−0.23 SDs, small effect size) compared to women in the “incomplete” group (Figure 3A, details in Appendix A). The same holds for the **RA sub-group** (PCS_intensity_: −0.15 SDs, PCS_interference_: −0.34 SDs, and PCS_side_: −0.27 SDs) (see Figure 3B and details in Appendix A). Except for PCS_intensity_, the RA cohort’s *p*-values were < 0.05 after Bonferroni-Holm correction.

Results for the **intra-operative** care group follow a similar pattern but are generally less pronounced. See Figure 3A for the total cohort and Figure 3B for the RA sub-group (details in Appendix A).

#### 3.5.3. Additional Associations with PCS_total_

In the total cohort, GA was associated with a higher PCS**_total_** compared to RA, and this was a trivial to small effect size (Table 3A). In both the total and RA cohorts, chronic pain and the administration of systemic opioids on the ward were associated with higher PCS_total_ (small effect size).

In the RA group, administration of neuraxial morphine was associated with a lower PCS**_total_**. This was a trivial to small effect size. It was associated with an improvement in the sub-scores for pain intensity and interference (both small effect sizes) and a worsening of the side-effects sub-score (small effect size; details in Appendix A).

## 4. Discussion

In this study, we evaluated a large cohort of women on the first day after CS. In the complete cohort, only 20% of women received all three guideline-recommended care elements (“full” care), and in the RA sub-group, it was 21%. Receiving all three elements of care was associated with a better PCS_total_ in both cases when compared to women who did not receive such care (“incomplete”). The sub-scores demonstrated that the benefit of this form of care was associated mainly with less interference with pain and fewer side-effects. The PCS, summarizing pain intensity, pain-related interference, and side effects, and the sub-scores, are an innovative tool offering a holistic view of the painful experience [30,31].

### 4.1. Individual PROs in “Full” versus “Incomplete” Care Groups

The association between “full” care and individual PROs was not uniform. For example, women receiving “full” care spent less time in severe pain, and pain interfered less with their sleep compared to those receiving “incomplete” care, yet their “worst pain since surgery” did not differ. “Time in Severe Pain” requests patients to reflect on how they experienced pain over the entire post-operative day. It may be a more appropriate measure for patients and clinicians compared to “worst pain,” which assesses a momentary event. In a study of patients undergoing mixed surgical procedures, “time in severe pain” was a risk factor for developing chronic pain, whereas “worst pain” was not [32]. Function, such as sleep, is a key feature in recovery after CS [4]. Despite the improvement in outcomes for “full-care” treated women, a high proportion of women still reported poor outcomes in the different domains.

### 4.2. Evidence about the Treatment Elements We Evaluated

Neuraxial anesthesia with an opioid is standard care for CS [33,34]. In the current cohort, this technique was employed by 77% of women. Spinal morphine is recommended and in doses of 50–100 μg, 9.7–26.6 h of analgesia is expected [5,35]. In the current cohort, only 18% of women received spinal morphine. The majority of women received the much shorter-acting drug fentanyl. We suggest that by the time women filled in the questionnaire (a median of 23:00 h after surgery), the effect of the RA block had worn off. This was reflected in the severe pain and high levels of interference that women reported, and up to 46% would have liked to receive more treatment for pain. Though, the Pain Composite Score_total_ for GA was higher than for RA, this was trivial to small effect size, indicating that beyond the first few hours after surgery, differences in outcomes between the two forms of anesthesia may be minor.

A full daily dose of at least one non-opioid analgesic was the second element of care we assessed. Scheduled full daily doses of paracetamol and NSAIDs are strongly recommended as a key component of multimodal analgesia after CS, based on high-quality procedure-specific evidence [6,36]. Two randomized controlled studies indicated that paracetamol, as a sole non-opioid analgesic for CS, did not reduce opioid consumption or pain scores [37,38]. Our findings indicate that 80% of women did not receive a full daily dose of a non-opioid analgesic. Hypertension or pre-eclampsia are relative contraindications for NSAID administration [39]. However, this limitation would apply to only a small proportion of the cohort. As parturients are generally healthy young women and need to care for the new-born, it is not justifiable to withhold this simple and inexpensive element of care, which might improve analgesia and recovery.

The third element of care we evaluated was the assessment of pain by care providers. Pain assessments have been under intense scrutiny; they are regarded as a “regulatory nuisance” [40]. Yet, due to the considerable variability in patients’ responses to pain and to analgesics, assessment, whatever form it takes, is the primary means for tailoring care to individual patients so that it might be effective and safe [10,41]. In the current cohort, pain was assessed in 43% of women. However, this might not have been clinically effective, as indicated by the high pain scores women reported, inconsistent analgesic treatment, and the fact that 40% of women reporting severe pain did not receive an opioid.

### 4.3. Factors Associated with a Higher Pain Composite Score (PCStotal)

Pre-existing chronic pain was associated with a worse PCS _total_. The magnitude of the effect was comparable to the protective effect of neuraxial morphine. Little is known about the prevalence of pre-existing chronic pain disorders in pregnant women [42]. In the current cohort, 7.5% of women reported that they experienced chronic pain lasting at least 3 months before CS, and the majority of women attributed this pain to the site of surgery. This might have been related to previous surgery, such as CS [43], or conditions such as low back and pelvic pain, hip or foot pain [42]. Receiving an opioid on the ward was another factor associated with a higher PCS. Due to the observational design of the study, causality cannot be determined. We suggest that women reporting severe pain receive an opioid.

Country income level did not have a significant effect on the PCS, though the sensitivity analysis revealed a trend for higher scores in women in high-income countries. An earlier PAIN OUT study evaluating PROs from 16,868 patients undergoing mixed surgical procedures and treated in 11 countries demonstrated that 94.3% of the variance was explained at the patient level, whereas “country” explained only 0.8% [44]. Thus, based on the current knowledge base, we suggest that a patient’s country of origin may not play a large role in determining the response to pain.

### 4.4. Strategies of Evaluating Care and Effect on Outcomes

In this study, we used an “all-versus-none” measurement strategy to evaluate the relationship between three measures of care and their effect on outcome [45]. This strategy forms the underlying premise of “care bundles,” which are a small set of evidence-based interventions for a defined patient population and setting that are generally accepted as elements of care that should be delivered to all patients [46]. When a “bundle” is implemented, outcomes tend to be better compared to when elements are implemented individually or not at all [47]. Thus, our findings lay the foundation for testing the effectiveness of a “bundle” approach for pain management after CS. We are unaware of other studies assessing the association between implementing a bundle-like approach for CS and pain-related PROs.

### 4.5. Strengths and Limitations

We had access to a large, international cohort of women undergoing CS, whose pain and treatments were assessed using a standardized methodology. PAIN OUT obtains core data about a large variety of surgical procedures, yet it does not address items that are specialized for specific interventions. We, thus, lack information about the urgency of CS, the surgical techniques employed, or the newborn. Due to the nature of data collection, we were unable to determine the reasons why medications, such as NSAIDs, were not administered. As the focus of PAIN OUT is on perioperative pain care we lack information related to surgical outcomes, such as time to urinary catheter removal or complications. In addition, the registry lacks information on pharmacoeconomic aspects of care, such as length of stay and re-admission rates. The latter are often country-specific and are likely to be unrelated to the management of pain on the first post-operative day. For evaluating the quality of care, as was the goal of this study, an assessment carried out once for each patient is sufficient [48]. Such a practice allowed us to obtain data from a large sample. A longitudinal evaluation aims to improve understanding of pain mechanisms. However, it significantly increases the complexity of the work, burdening both staff and patients [49]. We evaluated data spanning across a 10-year period, during which practices in some hospitals may have changed. We suggest that this does not invalidate our findings, as our primary aim was to demonstrate that combining recommended elements of care is associated with improved outcomes, as has been demonstrated in other fields of medicine [47,50,51,52]. A major motivation for participating in PAIN OUT is that collaborators are interested in evaluating care and carrying out quality improvement in their center. Thus, the findings may reflect practices in hospitals where they are at their best rather than being representative of care in any particular country.

## 5. Conclusions

Our findings point toward the benefits of offering care in such a manner that all recommended treatment elements are administered to a particular patient. Only a fifth of the current cohort received such care, yet this was associated with an improvement in outcomes and one that had a small to medium effect size. However, even in this group, a high proportion of women still reported poor outcomes. Thus, closing the evidence-practice gap and being in a situation where women after CS can be comfortable when caring for themselves and their newborn still remains a challenge.

## Figures and Tables

**Figure 1 jcm-12-00676-f001:**
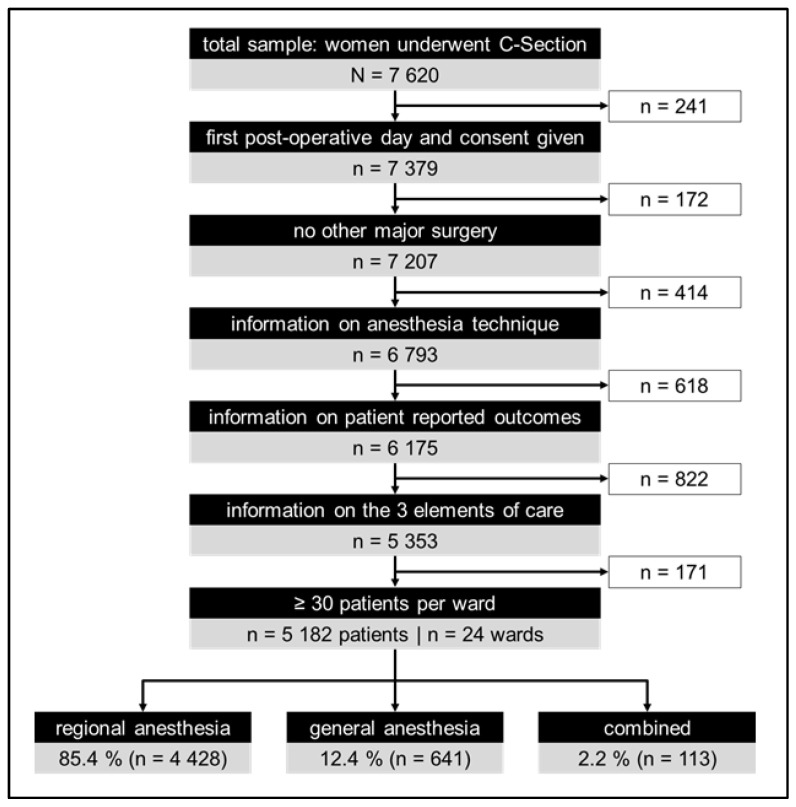
Study flow chart.

**Figure 2 jcm-12-00676-f002:**
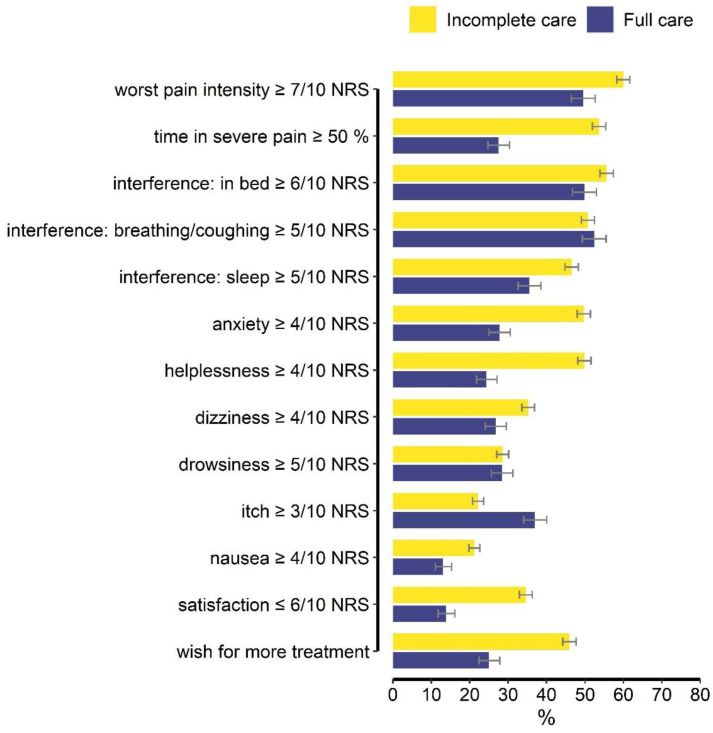
Presents a summary of dichotomized patient-reported outcome measures that form the Pain Composite Score and two additional outcomes. Results are shown separately for women who received “full” care (blue bars) compared to “incomplete” care (yellow bars). The capped lines indicate the 95% confidence intervals. ‘Satisfaction with results of pain care’ and ‘Would have liked more pain treatment’ are not part of the Pain Composite Score but are included here to provide additional information about women’s responses to the surgery and treatment.

**Figure 3 jcm-12-00676-f003:**
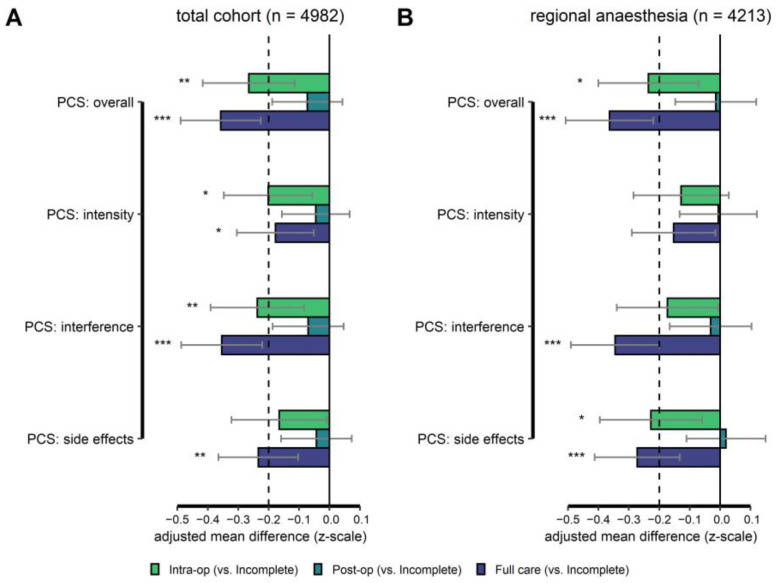
**Results of the regression analyses.** Regression weights (differences in covariate adjusted means), with 95% confidence intervals for each care group The “incomplete” group served as a reference. Significant differences between the “incomplete” group and other groups (intra-operative [green bars], post-operative [petrol bars], and full care [purple bars]) are marked with asterisks. All *p*-values were corrected using the Bonferroni-Holm procedure. The Pain Composite Score (and sub-scores) were z-standardized before modeling. Thus, mean differences can be interpreted in terms of standard deviations (absolute values ≥ 0.2 for small effect sizes, ≥ 0.5 for medium effect sizes, and ≥ 0.8 for large effect size). The broken vertical line at −0.2 level on the x-axis marks the level of a small effect size. Negative values indicate “better” outcomes compared to the “incomplete” group. Results for the total cohort are shown in (**A**) and results for women receiving only regional anesthesia in (**B**). * *p* < 0.05, ** *p* < 0.01, *** *p* < 0.001.

**Table 1 jcm-12-00676-t001:** Descriptive statistics of the cohort for continuous, dichotomous, and categorical variables.

Variable			Median	Q_1_	Q_3_	n_valid_
**Age** [years]			31.0	27.0	35.0	5 018
**Duration of surgery** [hh:mm]			0:50	0:37	1:05	5 114
**Time to survey** [hh:mm]			23:00	19:05	26:06	5 125
**Cumulative dose of non-opioid analgesics ^a^** [%]			93.5	39.4	177.3	5 182
**Total daily doses of opioids on the ward** [mg]						
*Morphine*		*i.m.*	15	10	15	789
		*s.c.*	12	6.5	16	176
		*p.o.*	15	15	15	123
*Pethidine*		*i.m.*	100	100	100	700
*Papaveretum*		*i.m.*	20	20	20	401
*Tramadol*		*p.o c.r.*	200	100	200	260
		*p.o.i.r.*	100	50	100	183
		*i.v.*	300	200	300	147
**Pain Composite Score-total (PCS_total_)** [0–10]	** *Total Cohort* **		3.5	2.4	4.8	5 182
*Regional Anesthesia*			3.6	2.4	4.8	4 428
*General Anesthesia*			3.3	2.3	4.4	641
*Combined*			3.7	2.7	5.2	113
**Variable**			**N**	**%**		**n_valid_**
**Chronic pain (≥3 months before surgery) ^b^**			385	7.5		5 145
**Intra-operative: neuraxial morphine**			922	17.8		5182
**Ward: non-opioid analgesic**		*Any*	4 509	87.0		5 182
*Paracetamol*			3 369	65.0		
*NSAID*			3 187	61.5		
*Metamizole*			344	6.6		
**Pain assessment by ward staff since return from surgery**		** *yes:* **	2210	42.6		5182
		** *no:* **	2972	57.4		
**Ward: systemic opioid**			3 108	60.0		5 182
**Ward: Patients reporting worst pain ≥ 7/10 NRS and received an opioid**			1160	39.1		2 968
**Peri-operative care groups**	** *Total cohort* **		**N**	**%**		**n_valid_**
**Incomplete care**		** *Cohort* **	3 251	62.7		5 182
*Regional anesthesia*			2 703	61.0		4 428
*General anesthesia*			485	75.7		641
*Combined RA & GA*			63	55.8		113
**Intra-operative care** [only]		** *Cohort* **	190	3.7		
*Regional anesthesia*			171	3.9		
*General anesthesia*			12	1.9		
*Combined RA & GA*			7	6.2		
**Post-operative care** [only]		** *Cohort* **	722	13.9		
*Regional anesthesia*			611	13.8		
*General anesthesia*			89	13.9		
*Combined RA & GA*			22	19.5		
**Full peri-operative** care		** *Cohort* **	1 019	19.7		
*Regional anesthesia*			943	21.3		
*General anesthesia*			55	8.6		
*Combined RA & GA*			21	18.6		

^a^ **The “sum of doses”** (intra- and post-operative) refers to full daily doses and the time between the end of surgery and the time of the survey. ^b^ In 26.4% of women, the **pre-existing chronic** pain was at the site of surgery, in 42% elsewhere, and in 32% at the site of surgery and elsewhere. i.m./s.c. = intramuscular/subcutaneous injection; p.o. = oral administration; c.r./i.r. = immediate/controlled release; i.v. = intravenous.

**Table 2 jcm-12-00676-t002:** Overview of opioids administered for neuraxial anesthesia (NA) The bold numbers listed in column “n” indicate the number of women who received morphine.

			Substances and Combinations		
	n	%	Morphine	Fentanyl	Sufentanil
	2735	52.8	No	yes	no
	**532**	10.3	Yes	yes	no
	285	5.5	No	no	yes
	**242**	4.7	Yes	no	yes
	**142**	2.8	Yes	no	no
	38	0.7	other opioid combinations		
	1212	23.4	no NA or with local anesthetics only		
**total:**	5182	100	922 (17.8%)	3275 (63.2%)	537 (10.4%)

**Table 3 jcm-12-00676-t003:** Results: the Pain Composite Score, total (PCS_total_), was used as the dependent variable in multivariable, multi-level regression models in (A) results for the total cohort and (B) for women with regional anesthesia. Standardized regression coefficients (β_z_) including 95% confidence intervals (95% CI) and *p*-values (*p*), are shown. Significant *p*-values are printed in bold.

		(A). Total Cohort [n = 4982]				(B). Regional Anesthesia (RA) [n = 4213]			
Variable	Reference	β_z_	95% CI		*p*	β_z_	95% CI		*p*
**(intercept)**		−0.27	−0.58	0.05	0.114	−0.30	−0.66	0.07	0.125
**Peri-operative care group:**									
Intra-operative [only]	[vs. Incomplete]	−0.27	−0.42	−0.11	**0.001**	−0.24	−0.40	−0.07	**0.005**
Post-operative [only]	[vs. Incomplete]	−0.07	−0.19	0.04	0.206	−0.01	−0.15	0.12	0.837
Full care	[vs. Incomplete]	−0.36	−0.49	−0.23	**<0.001**	−0.36	−0.51	−0.22	**<0.001**
**Anesthesia:**						*not modelled*			
General anesthesia	[vs. RA ]	0.16	0.04	0.28	**0.009**				
Combined RA & GA	[vs. RA]	0.04	−0.14	0.22	0.663				
**Age**	[years]	0.00	−0.01	0.00	0.140	0.00	−0.01	0.00	0.266
**Pre-existing chronic pain**	[vs. no]	0.21	0.11	0.31	**<0.001**	0.17	0.07	0.28	**0.002**
**Intra-operative: neuraxial morphine**	[vs. others/none]	*not modelled*				−0.16	−0.27	−0.05	**0.004**
**Ward: opioid**	[vs. no]	0.17	0.10	0.25	**<0.001**	0.20	0.12	0.28	**<0.001**
**Income level: high**	[vs. others]	0.32	−0.06	0.70	0.119	0.31	−0.11	0.73	0.162

R^2^ (total cohort, marginal, and conditional): 0.025 and 0.22. R^2^ (regional anesthesia, marginal, and conditional): 0.04 and 0.25.

## Data Availability

The raw data used and analyzed for the current study is available from the corresponding author on reasonable request.

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
