# Peer review of "Following Evidence-Based Recommendations for Perioperative Pain Management after Cesarean Section Is Associated with Better Pain-Related Outcomes: Analysis of Registry Dataâ€"

_jcm, 2023, doi:10.3390/jcm12020676_

Round 1
Reviewer 1 Report (Previous Reviewer 4)
Manuscript improved.
Author Response
Please see the attached file.
Reviewer 2 Report (Previous Reviewer 3)
Despite some noticeable improvements, I am afraid that my main concerns about the definition of subgroups have not been addressed, i.e. it is not possible to see what in the “full intervention” the main factor of analgesic efficacy was.
Author Response
Please see the attached file.

Reviewer 3 Report (Previous Reviewer 2)
The authors have addressed reviewers' comments and suggestions. I have no further comments.
Author Response
Please see the attached file.

Reviewer 4 Report (Previous Reviewer 1)
Well written article.
The authors have done a good job.
Congratulations to the authors for their good and important work.
Round 2
Reviewer 2 Report (Previous Reviewer 3)
Despite some noticeable improvements, I am afraid that my main concerns about the definition of subgroups have not been addressed, i.e. it is not possible to see what in the “full intervention” the main factor of analgesic efficacy was.
Author Response
Please check the attachment.

This manuscript is a resubmission of an earlier submission. The following is a list of the peer review reports and author responses from that submission.
Round 1
Reviewer 1 Report
Well written article. The complete article is well balanced.
Effective analgesia has a potential to improve postoperative recovery and outcomes, and has long-lasting positive effects on functional capacity and quality of life of patients. It is necessary to improve perioperative pain management in routine clinical practice.
The results of this study clearly demonstrated the importance of optimal utilization of the already available drugs and clinically proven techniques.
Congratulations for good work.
Reviewer 2 Report
The authors have conducted a retrospective cohort study based on PAIN OUT registry to assess early recovery outcomes after cesarean section. The main findings were that 1) in only 20% of parturients perioperative care followed evidence-based recommendations and 2) that early recovery was more favourable in parturients who were treated according to recommendations compared to those parturients who received care that was not compliant with recommendations.
The authors address an important clinical problem since cesarean section is a common surgical procedure worldwide and there appears to be no previous data on how compliance to perioperative care recommendations may affect outcomes. The manuscript is clear and most tables and figures are easy to understand. The ethics statements are adequate.
Strengths of this study were firstly, large sample size and secondly, that multi-dimensional patient-reported outcomes were evaluated.
I have only a few comments:
1) On table 1, the first perioperative care group is titled “Missing”. I suppose this is the ‘incomplete care’ group. If so, please correct it so that titles of perioperative groups are uniform troughout the manuscript.
2) Are the authors aware of any other data on how compliance to international guidelines or implementation of perioperative care protocols may affect outcomes after cesarean section? If such data does not seem to exist it should be stated in the discussion and if it does, the authors should compare their results to that data.
3) Please fill the data availability statement.
Reviewer 3 Report
Being aware of the different researches conducted out of the PAIN OUT registry, I will not challenge the quality of organisation, clinical setting, data collection and presentation. Nevertheless, according to my personal knowledge in postoperative analgesia – and particularly after caesarean section (CS) – I have to raise an important concern about how you defined your study subgroups. To me, the optimal protocol (i.e. what can be considered as a “full care”) for a CS is
1. in case of neuraxial anaesthesia to conduct the CS: perispinal morphine, followed by a convenient postoperative supplemental analgesia, which lies on NSAID/metamizol (± paracetamol), although a TAP or a wound infiltration on top of the neuraxial anaesthesia can also be a good alternative, at least for the 24 first postoperative hours;
2. in case of general anaesthesia or neuraxial anaesthesia without morphine, a proper postoperative analgesia must be achieved, and for this, it seems that at least two of these three interventions must be applied: TAP block, NSAID/metamizol, and systemic opioid.
Other protocols in which one of those major analgesic interventions misses cannot reach the same analgesic efficacy. This is particularly true for paracetamol alone (which mostly potentiates the other drugs) and for perispinal fentanyl/sufentanil (which were primarily used to potentiate anaesthesia and have anyway a short duration of action).
The other point is that you consider the perispinal opioid as “intraoperative care” while perispinal morphine is definitely an anticipated postoperative care, regarding the time frame of its analgesic action. On the other hand, as general anaesthesia is generally not an alternative to neuraxial anaesthesia (it is usually guided by obstetrical considerations), it does not make much sense to pool the two types of anaesthesia (although I admit that statistical power would be low in the general anaesthesia subgroup). Finally, studying the effect of the type of anaesthesia is an interesting endpoint, considering the potential preemptive effects of neuraxial anaesthesia.
I know this would lead to reconsider completely your main analyses, but I honestly believe that in the current configuration we miss a lot of clinical relevance and of new information.
My other comments are of less importance:
· If your wording “Pain Composite Score” has already been used in other studies, I admit that it would be hard to change, but the side effects of analgesics (mostly opioids) are more related to the treatment than to pain itself. For example, when you inject morphine intrathecally to all your patients, the resultant itch do not depend on pain intensity, as it was a preventive treatment.
· Could you please explain how you calculate your “adjusted PCS” (Z-score)? I am accustomed to Z-scores in case-control studies, but here what was your referential? Also, how must we interpret the Figure 2? I understand that non-overlapping whiskers signal a significant difference, but is the null value also a reference value to consider?
· There was no case treated by nefopam: is this amazing or not?
· It must be reminded that metamizol has been banned in many countries because of its severe haematological side effects.
Reviewer 4 Report
The authors investigated whether adherence to evidence-based recommendations for post-operative pain management after cesarean section was associated with a better outcome. They included a large sample of 50182 from the PAIN OUT registry over a time span of 10 years.
The manuscript clear, relevant for the field and presented in a well-structured manner and the citations are up to date. Hypothesis and statistics are sound.
Several results are somewhat surprising:
The incidence of chronic pain >3 three month before surgery with 7,5% was rather high. Even though this point is covered in the discussion it would be helpful to provide more information and to discuss further.
Although the three factors analyzed are common sense in the care of perioperative care after Cesarean section, surprisingly only 20% of the women were treated adequately.
This is a strong indication, that adherence to simple guidelines need to be improved.
Transversus abdominis plane (TAP) block (or local infiltration) was supposed to be added if general anesthesia was needed.
Dou you have data on the proportion of women who received a TAP-block after general anesthesia? TAP-block is an effective tool for postoperative pain control in these patients and it would be interesting to know if this regional anesthesia is used widely.
The results are clear, the conclusions are drawn properly.